# fMRI insights into differential brain activation, executive function, and physical activity in older adults

Huiqi Song[1], Jie Feng[2], Yingying Wang[3], Qichen Zhou[3], Chenglin Zhou[3], Jing Jin[3,4]*

**1** The Jockey Club School of Public Health and Primary Care, The Chinese University of Hong Kong, Hong Kong, China, **2** Department of Sports Science and Physical Education, The Chinese University of Hong Kong, Hong Kong, China, **3** School of Psychology, Shanghai University of Sport, Shanghai, China, **4** Key Laboratory of Exercise and Health Sciences of Ministry of Education, Shanghai University of Sport, Shanghai, China

* jinjing@sus.edu.cn

## Abstract

### Background

Executive function is vital for cognitive health, particularly in older adults, where declines can lead to an increased risk of cognitive impairment. Physical activity (PA) has been linked to improvements in executive function, yet the underlying mechanisms remain poorly understood.

### Methods

This cross-sectional study involved 41 Chinese adults (21 young: 23.0±2.12 years; 20 older: 63.30±2.36 years) who were categorized as physically active (≥3000 metabolic equivalent (MET)-min/week) or inactive (<3000 MET-min/week). Participants performed fMRI while completing executive function tasks (Flanker, N-back, Switching). Brain activation patterns were analyzed using Statistical Parametric Mapping (SPM), with significance thresholds set at $p < 0.01$ (voxel-level) and $p < 0.05$ (whole-brain corrected).

### Results

Physically active older adults showed significantly better accuracy and faster reaction times on the Flanker task than inactive peers. In young adults, those who were inactive exhibited greater activation in prefrontal regions during executive tasks. No significant differences in brain activation were found in older adults for these tasks. Additionally, activation in the right medial/paracentral cingulate gyrus (BA 6) negatively correlated with working memory reaction times in active young adults (r= −0.804, $p < 0.05$), whereas cognitive flexibility in active older adults positively correlated with activation in the right dorsolateral frontal gyrus (BA 32; r = 0.589, $p < 0.05$).

**Data availability statement:** All relevant data are within the paper and its Supporting Information files

**Funding:** This work was supported by Shanghai Science and Technology Commission (No. 17080503100) and National Natural Science Foundation of China (No. 31971024).

**Competing interests:** The authors have declared that no competing interests exist.

## Conclusion

Active older adults require less brain activation to perform executive function tasks, suggesting enhanced cognitive efficiency. In contrast, young adults showed different patterns of brain activation, indicating potential compensatory mechanisms. These results underscore PA's role in optimizing age-specific cognitive strategies and underscore the need for longitudinal research to clarify causality.

## Introduction

Advancements in technology and healthcare have led to a significant increase in the aging population. According to demographic forecasts, by 2030, roughly one in six people globally will be aged 60 or older [1]. The total number of older adults is expected to rise from 1 billion in 2020 to 1.4 billion by 2030, and further to 2.1 billion by 2050 [1]. This demographic transformation has intensified age-related public health challenges, particularly the decline in cognitive functions such as executive function, which is strongly associated with an increased risk of mild cognitive impairment and dementia [2], as well as deterioration in brain structure and function [3].

Executive function comprises a set of higher-order cognitive processes that support psychological well-being, academic performance, lifetime achievements, and multidimensional development across cognitive, social, and psychological domains [4]. Extensive empirical research conducted over several decades has established a consensus that executive function comprises three core fundamental components: inhibitory control, working memory, and cognitive flexibility [5,6]. These interconnected components serve distinct yet complementary functions: inhibitory control suppresses irrelevant stimuli, working memory stores and manipulates information, and cognitive flexibility adapts to shifting demands [4,7]. Given the profound implications of executive function decline on daily functioning and independence [8], identifying strategies to mitigate these effects has become a crucial area of research.

Physical activity (PA) has consistently shown beneficial effects on executive function across various population groups [9–11] and has the potential to alleviate age-related declines [12]. The connection between PA and executive function has been thoroughly investigated using diverse methodological approaches, including both observational and intervention studies [13,14]. A systematic review focusing on adults aged 50 and older, which analyzed 39 randomized controlled trials, found significant improvements in executive function resulting from PA interventions, irrespective of participants' initial cognitive status [14]. Similarly, a review of 25 randomized controlled trials involving adults over 60 years old found that PA positively enhanced inhibitory control, working memory, and cognitive flexibility [15]. Additionally, another systematic review indicated that PA may help slow cognitive decline in individuals with mild cognitive impairment [16].

However, the mechanism that explain the relationship between PA and executive function is still not well understood, particularly regarding how PA influences functional connectivity and neural efficiency. Previous research has utilized

approaches such as event-related potentials (ERPs) and functional near-infrared spectroscopy (fNIRS). For instance, one study found that older adults with higher levels of PA exhibited increased P3 and N1 amplitudes along with reduced P3 latency [17]. Another study indicated that physically active older adults experienced greater increases in prefrontal cortex oxygenation during tasks that demanded executive function compared to their inactive counterparts [18]. While these findings provide valuable insights, they are limited in their ability to comprehensively map whole-brain mechanisms or clarify how PA enhances functional connectivity and neuroplasticity across distributed neural networks.

Functional magnetic resonance imaging (fMRI) is often considered as the gold standard for the assessment of brain activity, and offers superior spatial resolution compared to methodologies such as ERPs and fNIRS [19], enabling precise localization of neural activity across distributed brain networks critical for executive function. Unlike ERPs, which excel in temporal resolution but lack spatial precision, and fNIRS, which is limited to cortical surface measurements, fMRI provides comprehensive mapping of both localized and network-level brain activity. This capability allows for the assessment of task-evoked activation and resting-state connectivity, offering unique insights into how PA modulates neural efficiency, plasticity, and compensatory mechanisms, particularly in aging populations. However, prior research on PA and executive function has predominantly relied on ERPs and fNIRS [20,21], which are constrained in their ability to examine whole-brain neural mechanisms. Additionally, many studies have focused on younger adults or lacked the spatial resolution necessary to investigate age-related neural adaptations. fMRI addresses these limitations by providing a robust framework to explore PA-related neural changes across the lifespan, filling critical gaps in understanding how PA influences executive function in older adults. To fill in the gap, this study aimed to examine executive function performance (accuracy and reaction time) across age groups (young versus older adults) and PA levels (high versus low); and (2) explore the association between brain activation patterns and performance in inhibitory control, working memory, and cognitive flexibility tasks.

## Methods

### Study design and participants

This cross-sectional study was conducted in Liaoning Province, China. Participants were recruited from the community centers using a snowball sampling approach. The inclusion criteria were: (1) Han Chinese adults aged 18 years or older; (2) had more than 9 years of education; (3) were at low risk of cognitive decline. The Montreal Cognitive Assessment Basic (MOCA-B) was used for cognitive screening. MOCA-B score ≥19 was regarded as at low risk for cognitive decline [22,23]; (4) without hearing, vision, or communication impairment; and (5) without physical or mental illness. Participants were excluded if they: (1) showed significant cognitive decline; (2) had a lifetime history of psychiatric disorders with psychotic features (psychosis, schizophrenia, or bipolar affective disorder); (3) were undergoing chemotherapy/radiation therapy; or (4) self-reported or exhibited hearing, vision, or communication impairments during screening (e.g., inability to hear conversational speech or read printed instructions in 12-point font). A total of 53 participants were included, including 22 young adults and 31 older adults.

The patients/participants provided their written informed consent to participate in this study. Data collection was carried out from October 2019 to December 2019. Participants completed demographic information and physical activity questionnaires under the guidance of research assistants. The Strengthening the Reporting of Observational Studies in Epidemiology (STROBE) statement was used to guide the study implementation and results presentation. This study involved human participants and was approved by the Ethical Committee of Shanghai University of Sport (No. 2016013). All participants provided written informed consent. To ensure confidentiality, all collected data were anonymized by assigning unique ID codes, with personal identifiers stored separately in password-protected files. Electronic data were stored on secure institutional servers with firewall protection, while paper records were kept in locked cabinets accessible only to the research team.

 

## Measurements

**Physical activity.** The International Physical Activity Questionnaire-Short Form (IPAQ-SF) assessed participants' physical activity levels [24]. The IPAQ-SF records the activity of four intensity levels, including vigorous physical activity (VPA), moderate physical activity (MPA), walking, and sitting over the past 7 days. The continuous score of the IPAQ-SF can be calculated as MET intensity (walking: 3.3 METs, MPA: 4 METs, and VPA: 8 METs) and is expressed in METs-min/week [25]. Participants were classified into the physically active group with PA achieving ≥3000 MET-min/week and the physically inactive group with PA achieving <3000 MET-min/week. This threshold aligns with the high activity category defined in the IPAQ scoring protocol [24]. The higher threshold is selected based on evidence that found PA was positively associated with brain health in older adults [26]. The IPAQ-SF has good reliability and validity in Chinese adults [27].

**Executive function.** Participants underwent fMRI scanning while performing computerized behavioral executive function tasks: the Flanker task, N-back task, and Switching task. Participants were measured in the fixed order of Flanker task, N-back task, and Switching task. These tests were carried out individually in a quiet room, under the guidance of a well-trained research assistant. The total duration was about 30 minutes, including the practice time and a brief break.

Inhibitory control was measured using the Eriksen Flanker task [28]. The stimuli consist of five horizontally aligned arrows, which are categorized into congruent and incongruent conditions. In the congruence condition, the direction of the centre arrow is consistent with the direction of the surrounding arrows (e.g., <<<<<or>>>>>). In the incongruent condition, the direction of the central arrow is different from the direction of the surrounding arrows (e.g., <<><<or>><<>>). Participants were directed to promptly and precisely press a key corresponding to the central arrow's direction. The task used a completely randomised EVENT design. The formal task consisted of 100 trials, with 50 trials per condition. Each trial started with a 1000 ms arrow stimulus, followed by an empty screen lasting between 2000 ms to 6000 ms with 500 ms intervals. The task was performed using a fully randomised event-related design and took 414 seconds. The reaction time and accuracy (% correct) indicated inhibitory control and information processing, respectively [29].

Working memory was assessed using the N-back task [30]. Participants were presented with a continuous stream of stimuli and asked to use their index finger to press a specific key for targets and their middle finger for non-target stimuli. Targets were identified based on the n-back task, meaning that a stimulus is considered a target if it matches the stimulus presented in n-stimuli earlier. This study used 0-back, 1-back, and 2-back tasks. In the 0-back task, the target was the letter X. Participants were asked to press the key with their index finger as soon as possible each time this letter appeared on the screen and to press the key with their middle finger for all other letters. In the 1-back and 2-back tasks, the target was any letter identical to the one that appeared one trial back and two trials back, respectively. The stimuli, including targets and non-targets, were presented randomly. Each task consisted of four blocks, and the protocol consisted of 12. The duration of the stimulus presentation was 1000 ms, and the inter-stimulus interval was 2000 ms. The task took 443 seconds. Each task's (0-back, 1-back, and 2-back) reaction time and accuracy (% correct) were recorded.

Cognitive flexibility was measured with the switching task [31]. Participants completed a color-shape paradigm. The stimuli included two shapes (circle or square), in one of two colors (red or blue), displayed in the centre of the screen. In each trial, participants were given an instructional cue above the two shapes (the word "color" or "shape"). In the shape block, participants were asked to judge if the two shapes were the same. In the color block, participants were required to judge if the colors of the two shapes were the same. In the switch block, participants alternated between shape and color judgments. Depending upon the cue word, participants were required to press the key with the index or middle finger to indicate whether the stimulus was the same or different. Participants were asked to respond as quickly and accurately as possible. There were four runs, each with three task blocks. In the first run, blocks were presented in the order of shape, color, and switch. In the following runs, the order was randomized. Each block had eight randomly appearing trials, with two shapes shown simultaneously for 1500 ms in each trial, followed by a 1500 ms blank screen. Participants were allowed to press a key when the stimulus appeared. The time used for this task was 443 seconds. Reaction time and accuracy (% correct) were recorded for shape, color, and switch blocks.

Task stimuli were generated by E-prime software (Psychology Software Tools, Inc, Pittsburgh, PA) and projected to a mirror attached to the MRI head coil via an MRI-compatible projector. The stimulus presentation program recorded the response time and accuracy of participants' responses for each trial. Trials with reaction times exceeding 2,800 ms or less than 100 ms were excluded [32,33]. These thresholds were selected based on neurophysiological evidence for minimum visual processing time and maintaining consistency with established cognitive task parameters. Regarding the practice sessions, participants completed a single experimental block (20 trials per task) to familiarize themselves with the task requirements. The transition criterion to formal scanning required participants to demonstrate a minimum accuracy threshold of 80% during practice trials. Upon completing the practice session, qualified participants immediately proceeded to the fMRI scanning phase. Due to thorough practice sessions before the formal scan, no trials were excluded.

**Covariates.** Participants' demographic data, including age, gender, education level, height, and weight, were collected through a self-reported questionnaire. The body mass index (BMI) was computed by dividing the body weight by the square of height ($kg/m^2$).

## Imaging data acquisition and analysis

Imaging data was collected on the GE Discovery MR-750 3.0T scanner (GE Discovery MR-750 3.0T scanner, GE Medical Systems, Waukesha, WI, USA) at the Brain and Cognitive Neuroscience Research Center of Liaoning Normal University. During the scanning, participants could not carry metallic objects and minimize head movements. High-resolution three-dimensional T1-weighted structural images were collected using a fast spoiled gradient-recalled echo (FSPGR) sequence (repetition time = 8.156 ms, echo time = 3.18 ms, flip angle = 8°, resolution = $1 \times 1 \times 1 mm^3$ isotropic, field of view = $256 \times 256 mm^2$, 176 interleaved slices, slice thickness = 1 mm, slice gap = 1 mm). Functional images were obtained using a gradient echo-planar sequence (repetition time = 2000 ms, echo time = 30 ms, flip angle = 90°, resolution = $3.44 \times 3.44 \times 3.2 mm^3$ isotropic, field of view = $220 \times 220 mm^2$, 43 interleaved slices, slice thickness = 3.2 mm, slice gap = 0 mm) [34].

**Preprocessing.** Imaging data were preprocessed using the Data Processing Assistant for Resting-State fMRI (DPARSF) toolbox (http://rfmri.org/DPARSF) [35], including temporal layer and head movement correction. All resulting images were interpolated to $3 \times 3 \times 3 mm^3$ resolution for group analysis in Montreal Neurological Institute (MNI) standard space. Participants with head motion greater than 2 mm were excluded from the final analysis [36]. The resulting functional images were spatially smoothed via a Gaussian kernel of $6 \times 6 \times 6 mm^3$ full width at half maximum (FWHM).

**First-level analysis.** After preprocessing, the data were analyzed using Statistical Parametric Mapping (SPM8) (Wellcome Centre for Human Neuroimaging, London, UK, http://www.fil.ion.ucl.ac.uk/spm). The first level analysis used general linear models to construct each participant's stimulus matrix. This matrix was then utilized to derive parameter estimates β images corresponding to each stimulus condition. Subsequently, contrast images of interest for various tasks were calculated. Specifically, for the Flanker task, the contrast was defined as the incongruent condition minus the congruent condition; in the N-back task, contrasts were computed as 1-back minus 0-back and 2-back minus 0-back; and for the switching task, the contrast was the switch condition minus the color and shape conditions.

**Second-level analysis.** In the second level analysis, independent samples t-tests were used to calculate the between-group contrast values for the different PA levels in the groups between each task (physically inactive in young adults vs. physically active in young adults, physically inactive in older adults vs. physically active in older adults). The independent t-test was conducted for whole-brain activation within each group with 0 as the reference value. Multiple comparison correction for whole-brain data was applied using Gaussian random field (GRF) theory. A voxel-significance level was set at $p < 0.01$ a whole-brain corrected significance level was set at $p < 0.05$. These parameters were selected based on previous studies [37,38].

Regions of interest (ROIs) were generated with a 6 mm-radius sphere centered on the peak voxel coordinate within each activation cluster [39]. To further investigate the relationship between PA and brain activation, Pearson's correlation

coefficients were calculated between the behavioral metrics (reaction time, correctness) and the brain imaging metrics (contrast values for each condition) for each group (young adults, older adults). Statistical significance levels were set at $p < 0.05$, and marginal significance $p < 0.1$ [40].

### Statistical analyses

The continuous data variables were displayed as mean ± standard deviation (SD), while categorical data were presented as participant count and percentage. Following adjustment for sex and education level, a two-way ANOVA was performed to examine the impact of physical activity (PA) levels (low-to-moderate, high) and age groups (young adults, older adults) on cognitive function. A Bonferroni post-hoc test was conducted to assess differences. All statistical analyses were carried out using SPSS version 27.0, with a significance level set at 0.05.

## Results

### Descriptive statistics

Of the 53 enrolled participants, 41 were included in the data analysis, while 12 were excluded due to scan absence (n = 5), withdrawal (n = 5), or incomplete questionnaires (n = 2). Table 1 displays participant characteristics. The young adult group (n = 21; 23.0 ± 2.12 years, 42.9% female) comprised 52.4% physically active individuals, compared to 60.0% in the older adult group (n = 20; 63.30 ± 2.36 years, 45.0% female). Excluded participants were significantly older than included participants (64.3 ± 2.1 vs. 42.6 ± 20.5 years, $p < 0.001$), with no other demographic differences.

### Executive function and physical activity in young and older adults

**Flanker task.** As shown in Table 2, for older adults, the incongruent accuracy was significantly higher in the physically active group than in the physically inactive group ($p < 0.05$). The incongruent reaction time was significantly shorter in the active group than in the inactive group ($p < 0.05$). For both young and older adults, the inactive group showed higher congruent minus incongruent accuracy than the active group ($p < 0.05$). No significant difference was found in other measures ($p > 0.05$).

**N-back task.** There was no significant difference in the accuracy and reaction time for 0-back, 1-back, and 2-back between the active and inactive groups in the young and old adults.

**Switching task.** For older adults, color accuracy was significantly lower in the active group than in the inactive group ($p < 0.05$). There was no significant difference in other measures between the active and inactive groups in the young and old adults.

### Functional activations during executive function tasks

**Flanker task.** Table 3 presents the results of a whole-brain analysis in young and older adults with different PA levels during executive function tasks. During the Flanker task, the inactive group in young adults was significantly more activated in the left medial frontal cortex (BA 10), left dorsolateral frontal gyrus (BA 10), left middle frontal gyrus (BA 10), left middle temporal gyrus (BA 39), and left angular gyrus (BA 39) than the active group. There was no significant difference in brain activation between inactive and active groups in older adults.

Inactive young adults required broader activation across frontal and temporoparietal regions to perform the Flanker task compared to their active counterparts, while no such PA-related differences were observed in older adults.

**N-back task.** Among young adults, during the 0-back task, the activation in the bilateral medial and paracentral cingulate gyrus (BA 24) and right precentral gyrus (BA 6) of the inactive group was significantly higher than the active group. In 1-back tasks, the activation in the bilateral medial and paracentral cingulate gyrus (BA 24), right anterior precentral gyrus (BA 6), and right insula (BA 24) of the inactive group was significantly greater than the active group. In

**Table 1. Characteristics of participants.**

| Variables | Young adults (n=21) | Older adults (n=20) | P value[a] |
|---|---|---|---|
| Age, years | 23.0±2.12 | 63.30±2.36 | <0.001 |
| Age range, years | 19-27 | 59-69 | |
| Sex | | | |
| Male | 12 (57.1%) | 11 (55.0%) | 0.893 |
| Female | 9 (42.9%) | 9 (45.0%) | |
| Height, cm | 171.86±7.40 | 167.70±6.90 | 0.070 |
| Weight, kg | 65.52±11.17 | 72.98±10.00 | 0.030 |
| Body mass index, kg/m² | 22.10±2.91 | 25.87±2.55 | <0.001 |
| Educational attainment, years | 16.52±1.50 | 11.10±1.97 | <0.001 |
| Physical activity level, METs-min/week | 4411.31±3432.20 | 6826.91±7900.18 | 0.208 |
| Physical activity | | | |
| Physically inactive | 10 (47.6%) | 8 (40.0%) | 0.633 |
| Physically active | 11 (52.4%) | 12 (60.0%) | |
| Executive function | | | |
| Flanker task ACC, % | | | |
| Congruent | 0.99±0.03 | 0.97±0.07 | 0.168 |
| Incongruent | 0.98±0.03 | 0.90±0.20 | 0.065 |
| Congruent-Incongruent | 0.07±0.19 | 0.01±0.02 | 0.003 |
| Flanker task RT, ms | | | |
| Congruent | 546.86±57.92 | 644.10±88.39 | <0.001 |
| Incongruent | 630.63±61.85 | 742.71±128.53 | 0.001 |
| Congruent-Incongruent | −98.61±82.75 | −83.76±36.13 | 0.183 |
| N-back task ACC | | | |
| 0-back | 0.98±0.03 | 0.95±0.06 | 0.045 |
| 1-back | 0.95±0.07 | 0.77±0.19 | <0.001 |
| 2-back | 0.85±0.15 | 0.76±0.12 | 0.090 |
| N-back task RT, ms | | | |
| 0-back | 605.25±64.14 | 703.97±102.12 | 0.001 |
| 1-back | 663.26±70.80 | 796.89±131.78 | <0.001 |
| 2-back | 747.66±183.12 | 844.63±128.83 | 0.058 |
| Switching task ACC | | | |
| Shape | 0.98±0.03 | 0.95±0.06 | 0.045 |
| Color | 0.95±0.07 | 0.77±0.19 | <0.001 |
| Switch | 0.85±0.15 | 0.76±0.12 | 0.090 |
| Switching task RT, ms | | | |
| Shape | 605.25±64.14 | 703.97±102.12 | 0.001 |
| Color | 663.26±70.80 | 796.89±131.78 | <0.001 |
| Switch | 747.66±183.12 | 844.63±128.83 | 0.058 |

Note: Data are presented as mean ±SD or number (percentage).

[a]Independent sample t-test or Chi-square test were used to compare the means or proportion between young adults and older adults.

Abbreviation: ACC, accuracy; RT, reaction time.

**Table 2. Executive function performance by physical activity level in young and older adults.**

| Variables | Young adults (n=21) | | | | | Older adults (n=20) | | | | |
|---|---|---|---|---|---|---|---|---|---|---|
| | Physically inactive (n=10) | Physically active (n=11) | Mean difference (95% CI) | Effect size, Cohen's d | P value | Physically inactive (n=8) | Physically active (n=12) | Mean difference (95% CI) | Effect size, Cohen's d | P value |
| **Flanker task accuracy** | | | | | | | | | | |
| Congruent | 0.99±0.03 | 0.99±0.02 | 0.00 (−0.03, 0.02) | 0.18 | 0.683 | 0.97±0.04 | 0.96±0.08 | 0.01 (−0.06, 0.07) | 0.08 | 0.871 |
| Incongruent | 0.97±0.04 | 0.99±0.02 | −0.02 (−0.05, 0.01) | 0.71 | 0.123 | 0.78±0.28 | 0.97±0.05 | −0.20 (−0.36, −0.03) | 1.11 | 0.026 |
| Congruent-Incongruent | 0.02±0.02 | 0.00±0.01 | 0.02 (0.00, 0.03) | 1.20 | 0.019 | 0.19±0.26 | −0.01±0.04 | 0.20 (0.04, 0.36) | 1.20 | 0.017 |
| **Flanker task reaction time, ms** | | | | | | | | | | |
| Congruent | 528.72±63.09 | 563.35±49.99 | −34.63 (−86.37, 17.10) | 0.61 | 0.177 | 669.22±117.65 | 627.35±62.61 | 41.88 (−42.69, 126.45) | 0.48 | 0.312 |
| Incongruent | 629.96±78.01 | 631.23±46.62 | −1.27 (−59.30, 56.77) | 0.02 | 0.964 | 811.24±175.91 | 697.02±56.12 | 114.21 (0.92, 227.51) | 0.97 | 0.048 |
| Congruent-Incongruent | −101.24±40.11 | −67.87±24.07 | −33.37 (−63.24, −3.49) | 1.02 | 0.075 | −142.01±113.36 | −69.68±36.96 | −72.34 (−145.57, 0.89) | 0.95 | 0.530 |
| **N-back task accuracy** | | | | | | | | | | |
| 0-back | 0.98±0.04 | 0.98±0.03 | −0.01 (−0.03, 0.02) | 0.16 | 0.712 | 0.94±0.08 | 0.95±0.05 | −0.02 (−0.08, 0.04) | 0.31 | 0.506 |
| 1-back | 0.95±0.07 | 0.95±0.07 | 0.00 (−0.06, 0.07) | 0.01 | 0.979 | 0.80±0.12 | 0.75±0.23 | 0.04 (−0.14, 0.23) | 0.23 | 0.622 |
| 2-back | 0.88±0.16 | 0.81±0.14 | 0.07 (−0.06, 0.21) | 0.50 | 0.267 | 0.77±0.15 | 0.75±0.19 | 0.02 (−0.15, 0.19) | 0.12 | 0.799 |
| **N-back task reaction time, ms** | | | | | | | | | | |
| 0-back | 594.84±41.29 | 614.71±80.54 | −19.88 (−79.29, 39.54) | 0.31 | 0.492 | 687.15±102.72 | 715.19±104.66 | −28.05 (−127.69, 71.59) | 0.27 | 0.562 |
| 1-back | 635.61±57.41 | 688.40±74.83 | −52.79 (−114.19, 8.61) | 0.79 | 0.088 | 738.47±112.36 | 835.84±133.51 | −97.38 (−217.92, 23.17) | 0.78 | 0.107 |
| 2-back | 686.62±166.61 | 803.15±187.00 | −116.54 (−278.99, 45.91) | 0.66 | 0.150 | 811.99±73.92 | 866.39±154.59 | −54.40 (−178.43, 69.63) | 0.42 | 0.369 |
| **Switching task accuracy** | | | | | | | | | | |
| Shape | 0.97±0.03 | 0.96±0.05 | 0.01 (−0.03, 0.05) | 0.26 | 0.554 | 0.89±0.11 | 0.78±0.16 | 0.11 (−0.03, 0.25) | 0.77 | 0.107 |
| Color | 0.95±0.05 | 0.95±0.07 | 0.00 (−0.06, 0.05) | 0.01 | 0.974 | 0.97±0.05 | 0.90±0.07 | 0.06 (0.00, 0.13) | 1.02 | 0.038 |
| Switch | 0.94±0.06 | 0.95±0.04 | −0.01 (−0.05, 0.03) | 0.24 | 0.595 | 0.83±0.11 | 0.77±0.08 | 0.06 (−0.03, 0.15) | 0.66 | 0.167 |
| **Switching task reaction time, ms** | | | | | | | | | | |
| Shape | 894.08±102.36 | 917.42±140.60 | 23.34 (−136.71, 90.02) | 0.19 | 0.671 | 1106.88±100.47 | 1181.12±139.07 | −74.24 (−194.56, 46.09) | 0.59 | 0.211 |
| Color | 908.83±119.03 | 914.00±125.05 | −5.17 (−116.96, 106.62) | 0.04 | 0.924 | 1035.57±117.15 | 1240.32±145.24 | −56.01 (−161.72, 49.70) | 0.51 | 0.280 |
| Switch | 1013.22±126.23 | 996.40±157.83 | 16.82 (−114.62, 148.26) | 0.12 | 0.792 | 1240.32±145.24 | 1246.85±105.33 | −6.54 (−123.92, 110.84) | 0.05 | 0.908 |

**Table 3. Differences in brain region activation between the different PA levels during executive function tasks.**

| Task | Region | Brodmann area | Voxels | Peak T | Montreal Neurological Institute coordinates | | |
|---|---|---|---|---|---|---|---|
| | | | | | X | Y | Z |
| Flanker task | Young adults: physically inactive vs. physically active | | | | | | |
| | Left medial frontal cortex | 10 | 57 | −4.58 | −12 | 54 | 18 |
| | Left dorsolateral frontal gyrus | 10 | 46 | −4.58 | −12 | 54 | 18 |
| | Left middle frontal gyrus | 10 | 60 | −4.58 | −12 | 54 | 18 |
| | Left middle temporal gyrus | 39 | 84 | −4.32 | −42 | −54 | 18 |
| | Left angular gyrus | 39 | 40 | −4.32 | −42 | −54 | 18 |
| | Older adults: physically inactive vs. physically active | | | | | | |
| | Nil | | | | | | |
| N-back task | Young adults: physically inactive vs. physically active | | | | | | |
| 0-back | Right precentral gyrus | 6 | 80 | −4.82 | 18 | −9 | 39 |
| | Right medial and paracentral cingulate gyrus | 24 | 106 | −4.82 | 18 | −9 | 39 |
| | Left medial and paracentral cingulate gyrus | 24 | 77 | −4.82 | 18 | −9 | 39 |
| 1-back | Right precentral gyrus | 6 | 88 | −4.99 | 21 | −9 | 39 |
| | Right medial and paracentral cingulate gyrus | 24 | 86 | −4.99 | 21 | −9 | 39 |
| | Left medial and paracentral cingulate gyrus | 24 | 58 | −4.99 | 21 | −9 | 39 |
| | Right insula | 24 | 53 | −4.99 | 21 | −9 | 39 |
| 2-back | Right precentral gyrus | 6 | 46 | −4.31 | 27 | −9 | 42 |
| | Right medial and paracentral cingulate gyrus | 6 | 46 | −4.31 | 27 | −9 | 42 |
| | Older adults: physically inactive vs. physically active | | | | | | |
| 0-back | Left middle frontal gyrus | 46 | 96 | 5.49 | −15 | 12 | 24 |
| 1-back | Left middle frontal gyrus | 48 | 55 | 5.92 | −15 | 15 | 21 |
| 2-back | Left middle frontal gyrus | 48 | 59 | 6.12 | −15 | 18 | 18 |
| Switching task | Young adults: physically inactive vs. physically active | | | | | | |
| | Right supraorbital gyrus | 11 | 32 | −4.49 | 18 | 54 | −12 |
| | Right supraorbital frontal gyrus | 11 | 126 | −4.49 | 18 | 54 | −12 |
| | Right infraorbital gyrus | 11 | 51 | −4.49 | 18 | 54 | −12 |
| | Right lingual gyrus | 17 | 38 | −4.67 | 0 | −54 | −24 |
| | Cerebellar vermis | 17 | 45 | −4.67 | 0 | −54 | −24 |
| | Right medial and paracingulate gyrus | 32 | 62 | −4.43 | −9 | 33 | 24 |
| | Left medial frontal cortex | 32 | 34 | −4.43 | −9 | 33 | 24 |
| | Left anterior cingulate and paracingulate gyrus | 32 | 59 | −4.43 | −9 | 33 | 24 |
| | Older adults: physically inactive vs. physically active | | | | | | |
| | Right lingual gyrus | 17 | 34 | 4.08 | −33 | −81 | 3 |
| | Right pericalcarine cortex | 17 | 98 | 4.08 | −33 | −81 | 3 |
| | Left occipital gyrus | 19 | 49 | 4.08 | −33 | −81 | 3 |
| | Right dorsolateral frontal gyrus | 32 | 111 | 5.31 | 12 | 15 | 24 |
| | Right subfrontal gyrus | 32 | 55 | 5.31 | 12 | 15 | 24 |
| | Right anterior cingulate and paracingulate gyrus | 32 | 95 | 5.31 | 12 | 15 | 24 |
| | Right medial and paracingulate gyrus | 32 | 32 | 5.31 | 12 | 15 | 24 |
| | Right supplementary motor area | 32 | 70 | 5.31 | 12 | 15 | 24 |
| | Left medial frontal cortex | 32 | 56 | 5.31 | 12 | 15 | 24 |
| | Left dorsolateral frontal gyrus | 32 | 103 | 5.31 | 12 | 15 | 24 |
| | Left anterior cingulate and paracingulate gyrus | 32 | 35 | 5.31 | 12 | 15 | 24 |
| | Right middle frontal gyrus | 46 | 181 | 5.31 | 12 | 15 | 24 |
| | Left middle frontal gyrus | 46 | 120 | 5.31 | 12 | 15 | 24 |

Note: Activation of all listed voxels was statistically significant at a GRF corrected p < 0.01.

the 2-back task, the activation in the right precentral gyrus (BA 6) and right medial and paracentral cingulate gyrus (BA 6) of the inactive group was significantly greater compared to the active group. During 0-back, 1-back, and 2-back tasks, the activation in the left middle frontal gyrus (BA 46, BA 48) of the inactive group was significantly greater than the active group in older adults.Inactive participants in both age groups showed heightened activation in prefrontal and cingulate regions. This pattern was particularly pronounced in older adults during higher cognitive loads (2-back).

**Switching task.** The activation in the right supraorbital gyrus (BA 11), right supraorbital frontal gyrus (BA 11), right infraorbital gyrus (BA 11), right lingual gyrus (BA 17), cerebellar vermis (BA 17), right medial and paracingulate gyrus (BA 32), left medial frontal cortex (BA 32), and left anterior cingulate and paracingulate gyrus (BA 32) of the inactive group was significantly higher than the active group in young adults. Among older adults, the inactive group was found in the bilateral dorsolateral frontal gyrus (BA 32), bilateral anterior cingulate and paracingulate gyrus (BA 32), bilateral middle frontal gyrus (BA 46), right lingual gyrus (BA 17), right pericalcarine cortex (BA 17), left occipital gyrus (BA 19), right subfrontal gyrus(BA 32), right medial and paracingulate gyrus (BA 32), right supplementary motor area (BA 32), and left medial frontal cortex (BA 32) were significantly more activated than the active group.

Inactive individuals demonstrated widespread hyperactivation involving frontal, cingulate, and visual processing areas.

## Correlation between brain activation and executive function tasks

**Flanker task.** As shown in Fig 1, there was no significant correlation between brain activation and the Flanker task across different PA levels in young adults (p > 0.1). There was no correlation between brain activation and the Flanker task in older adults.

**N-back task.** As shown in Fig 2 and Fig 3, in young adults, the reaction time of the 2-back task was negatively correlated with the right medial and paracentral cingulate gyrus (BA 6) in the active group (r = −0.804, p = 0.003). There was no significant correlation between brain activation and other N-back task performance in young adults, and no significant correlation between brain activation and N-back task was found in older adults.

**Switching task.** Table 4 and Fig 4 show that no significant correlation between brain activation and the switching task was observed in young adults. Table 4 and Fig 5 show that there was a significant positive correlation between the right dorsolateral frontal gyrus in older adults (BA 32) and response time of the switching task in the active group (r = 0.589,

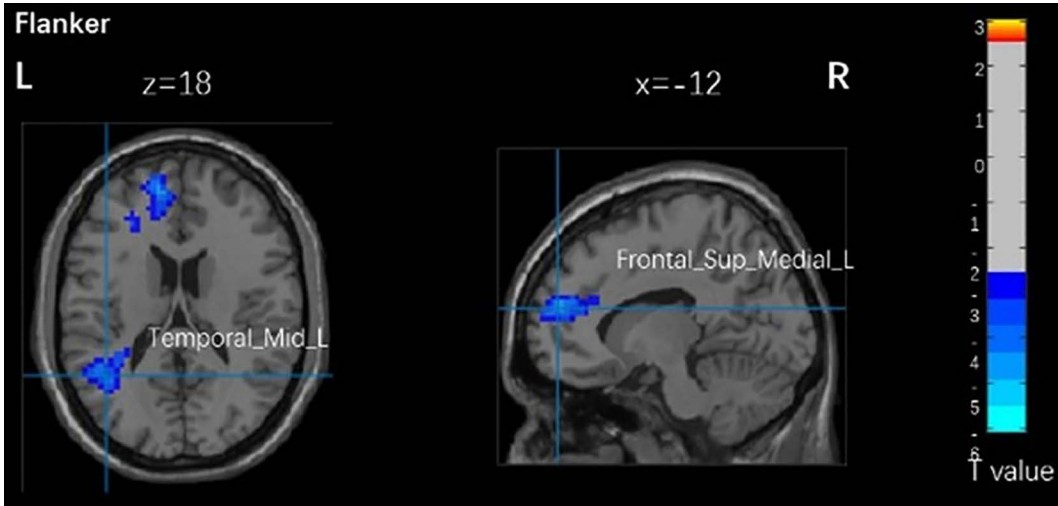

**Fig 1. Brain activation related to Flanker task brain areas with different physical activity levels in young adults.** Temporal_Mid_L, left middle temporal gyrus. Frontal_Sup_Medial_L, left superior medial frontal cortex.

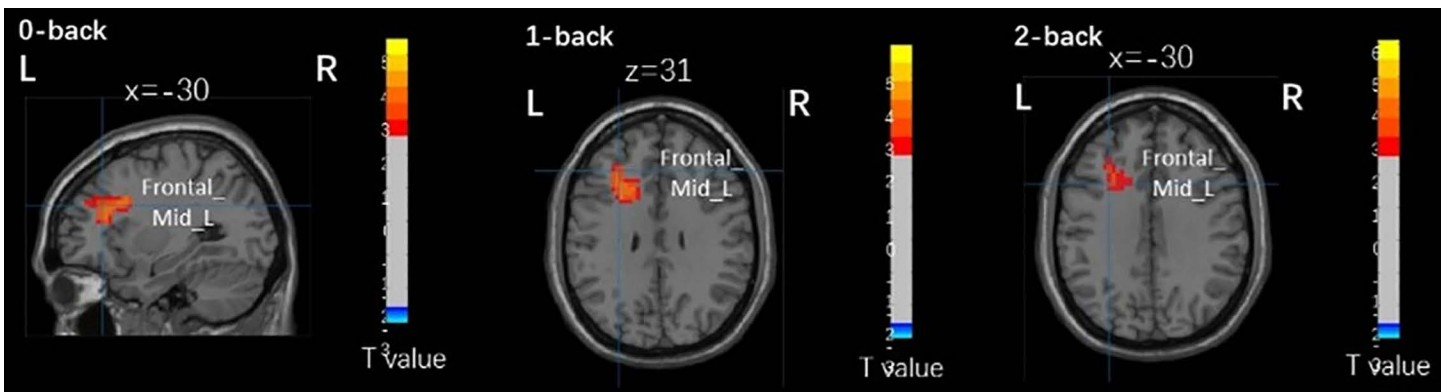

**Fig 2. Brain activation related to N-back task brain areas with different physical activity levels in young adults.** Cingulum_Mid_R, right medial and paracentral cingulate gyrus. Precentral_R, right precentral gyrus.

**Fig 3. Brain activation related to N-back task brain areas with different physical activity levels in older adults.** Frontal_Mid_L, left middle frontal gyrus.

**Table 4. Correlation between brain activation and executive function performance by physical activity level in young and older adults.**

| Task | Region | Physically inactive group | | | | Physically active group | | | |
|---|---|---|---|---|---|---|---|---|---|
| | | Accuracy | P value | Reaction time | P value | Accuracy | P value | Reaction time | P value |
| Flanker task | Young adults | | | | | | | | |
| | Left middle temporal gyrus | 0.409 | *0.241* | −0.266 | *0.457* | 0.354 | *0.285* | −0.160 | *0.638* |
| | Left medial frontal cortex | 0.041 | *0.911* | −0.043 | *0.906* | −0.163 | *0.632* | 0.320 | *0.338* |
| | Older adults | | | | | | | | |
| | Nil | | | | | | | | |
| N-back task | Young adults | | | | | | | | |
| 0-back | Right medial and paracentral cingulate gyrus | 0.194 | *0.591* | 0.111 | *0.760* | −0.389 | *0.266* | 0.246 | *0.493* |
| 1-back | Right medial and paracentral cingulate gyrus | −0.504 | *0.137* | −0.367 | *0.298* | −0.101 | *0.767* | −0.334 | *0.316* |
| | Right precentral gyrus | −0.405 | *0.246* | 0.132 | *0.715* | −0.184 | *0.588* | −0.064 | *0.853* |
| 2-back | Right medial and paracentral cingulate gyrus | −0.438 | *0.206* | −0.216 | *0.548* | 0.379 | *0.251* | −0.804 | *0.003* |
| | Right precentral gyrus | 0.346 | *0.327* | 0.129 | *0.723* | 0.049 | *0.886* | −0.222 | *0.512* |
| | Older adults | | | | | | | | |
| 0-back | Left middle frontal gyrus | 0.088 | *0.837* | −0.553 | *0.155* | −0.415 | *0.179* | −0.480 | *0.115* |
| 1-back | Left middle frontal gyrus | −0.554 | *0.154* | 0.535 | *0.172* | −0.114 | *0.725* | −0.152 | *0.636* |
| 2-back | Left middle frontal gyrus | −0.321 | *0.438* | 0.303 | *0.466* | −0.017 | *0.957* | −0.195 | *0.543* |
| Switching task | Young adults | | | | | | | | |
| | Cerebellar vermis | −0.059 | *0.871* | −0.096 | *0.792* | 0.344 | *0.300* | −0.130 | *0.703* |
| | Right lingual gyrus | 0.105 | *0.772* | −0.105 | *0.774* | 0.348 | *0.294* | −0.255 | *0.450* |
| | Right supraorbital gyrus | 0.308 | *0.387* | −0.218 | *0.545* | 0.199 | *0.558* | 0.329 | *0.323* |
| | Left anterior cingulate and paracingulate gyrus | −0.164 | *0.652* | −0.369 | *0.294* | 0.072 | *0.835* | −0.223 | *0.511* |
| | Older adults | | | | | | | | |
| | Right pericalcarine cortex | −0.531 | *0.175* | −0.028 | *0.947* | −0.138 | *0.669* | 0.347 | *0.270* |
| | Left occipital gyrus | −0.387 | *0.344* | 0.242 | *0.563* | −0.159 | *0.622* | 0.149 | *0.644* |
| | Right middle frontal gyrus | 0.016 | *0.971* | 0.151 | *0.721* | −0.484 | *0.111* | 0.488 | *0.107* |
| | Left middle frontal gyrus | −0.187 | *0.658* | 0.010 | *0.981* | −0.174 | *0.588* | 0.266 | *0.403* |
| | Right dorsolateral frontal gyrus | −0.323 | *0.435* | 0.583 | *0.129* | −0.328 | *0.298* | 0.589 | *0.044* |
| | Right anterior cingulate and paracingulate gyrus | −0.185 | *0.661* | 0.434 | *0.282* | −0.314 | *0.320* | 0.512 | *0.089* |

p = 0.044), the left anterior cingulate and paracingulate gyrus (BA 32) were positively correlated with response time in the active group (r = 0.512, p = 0.089). There was no significant correlation between brain activation and other switching task performance in older adults.

## Discussion

This study explored how PA affects brain activation about the decline of executive function (inhibitory control, working memory, cognitive flexibility). Compared to the physically active group, the physically inactive group exhibited greater activation across multiple brain regions during executive function tasks. Working memory reaction time was negatively correlated with activation in active young adults' right medial and paracentral cingulate gyrus (BA 6). Active older adults showed a significant positive correlation between the cognitive flexibility reaction time and activation in the right dorsolateral frontal gyrus (BA 32) and left anterior cingulate and paracingulate gyrus (BA 32).

During the Flanker task, the physically inactive group in young adults required more activation in the prefrontal brain regions (BA 10, BA 39) to accomplish the executive control task than the active group. This may reflect compensatory neural recruitment resulting from reduced cognitive efficiency. These regions are closely associated with higher-order

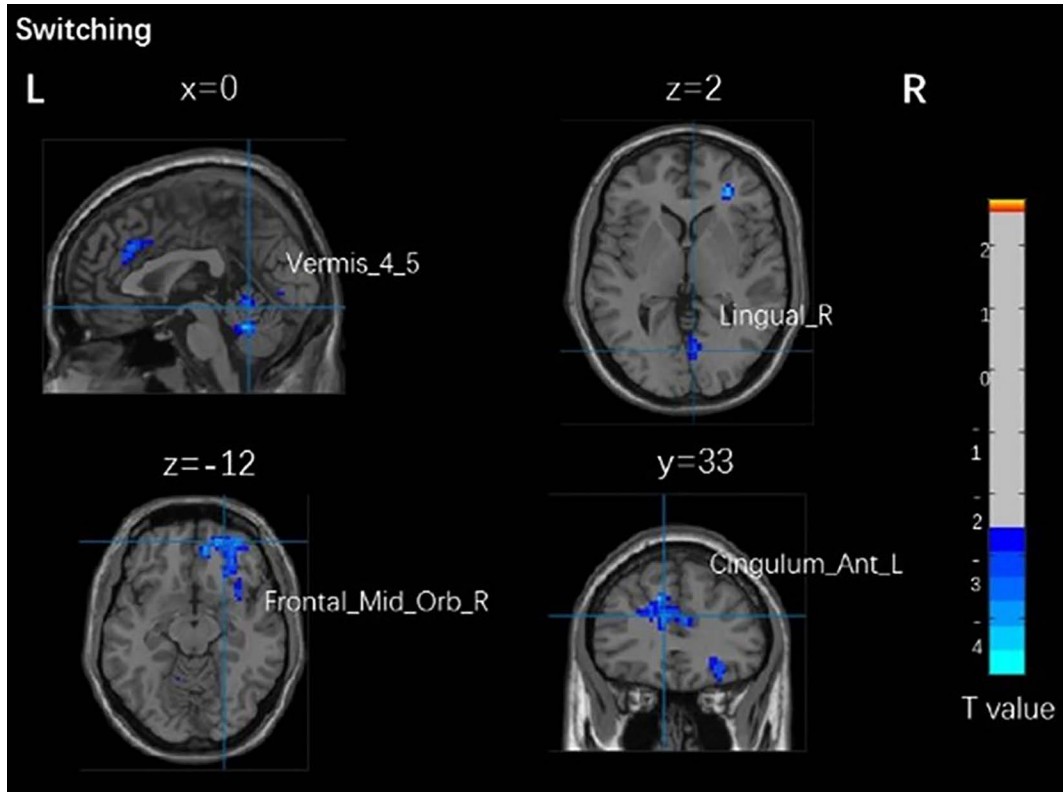

**Fig 4. Brain activation related to switching task brain areas with different physical activity levels in young adults.** Vermis_4_5, cerebellar vermis. Lingual_R, right lingual gyrus. Frontal_Mid_Orb_R, right supraorbital gyrus. Cingulum_Ant_L, left anterior cingulate and paracingulate gyrus.

cognitive functions such as attentional control, conflict monitoring, and semantic processing [41,42]. The result aligns with the neural inefficiency hypothesis, suggesting that physically active individuals can achieve similar performance with less neural resource expenditure [43,44]. In contrast, no significant differences in brain activation were observed between the physically inactive and active groups among older adults. This finding is consistent with the dedifferentiation hypothesis of executive function decline, which posits that aging reduces the brain's ability to selectively recruit task-specific neural resources, leading to broader, less efficient activation patterns [45]. For the N-back task, the findings demonstrate that PA does not reverse dedifferentiation in older adults but may instead promote compensatory recruitment of alternative neural circuits, particularly in the left middle frontal gyrus (BA 6, BA 24, BA 48), a region crucial for information maintenance, attentional control, and interference resolution [46]. This aligns with the scaffolding theory of aging and cognition [47], which posits that older adults rely on additional neural resources to maintain cognitive performance. The diffuse activation observed in our study suggests that PA may be associated with this compensatory scaffolding, particularly in open-skill exercises (e.g., tennis), which engage broader neural networks [48]. While PA may not restore youthful neural specificity, it can optimize the brain's ability to compensate for age-related declines, particularly in executive function. Future research should explore whether long-term PA can delay the onset of dedifferentiation or elevate its compensatory mechanisms.

Our study revealed that brain activation differences across PA levels were more pronounced in older adults, who recruited broader neural resources during the switching task compared to younger adults. This expanded activation pattern in older adults suggests an age-related shift toward more distributed neural processing to maintain cognitive performance. In addition, low-moderate PA levels required more prefrontal activation to achieve the cognitive flexibility

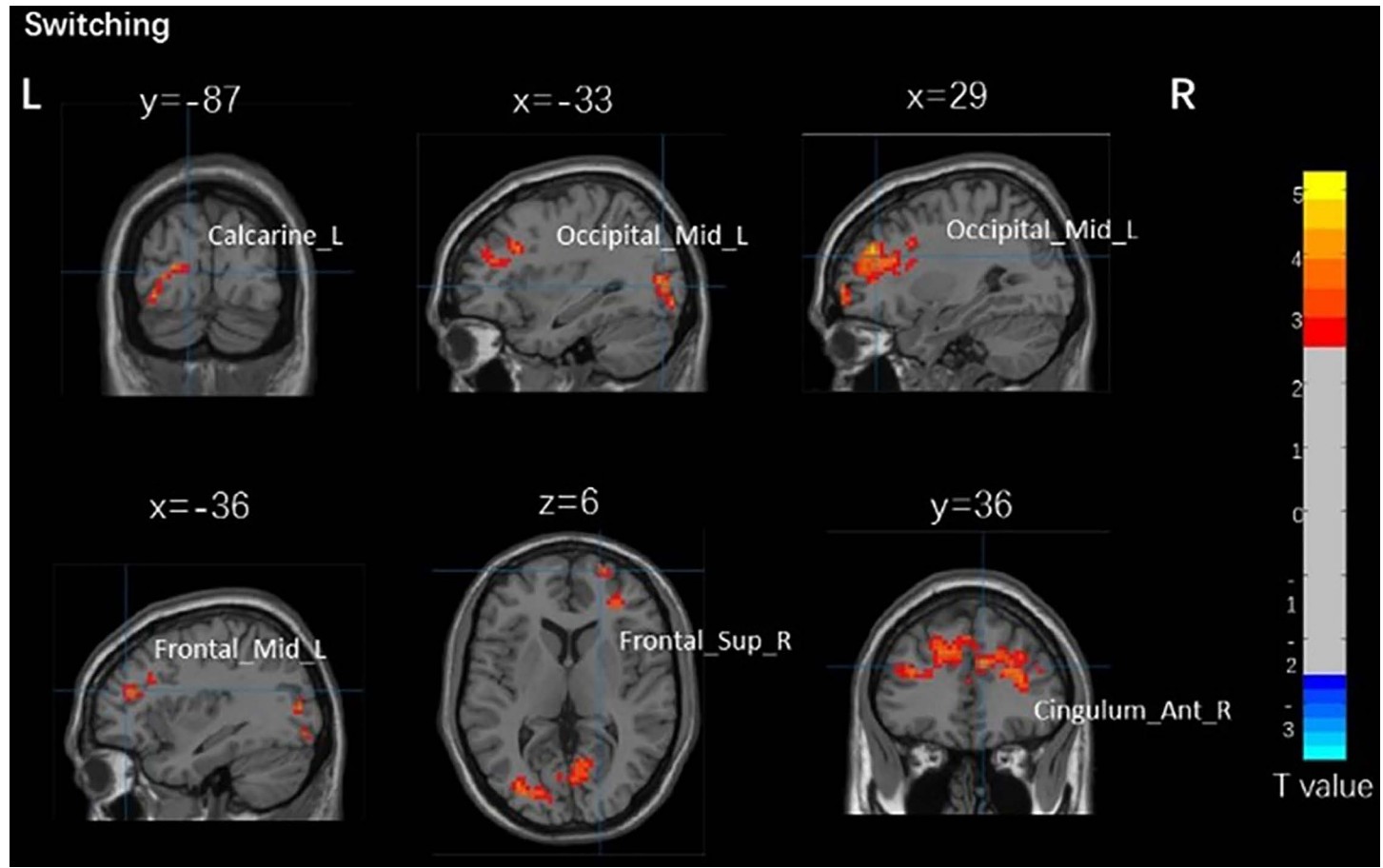

**Fig 5. Brain activation related to switching task brain areas with different physical activity levels in older adults.** Calcarine_R, right pericalcarine cortex. Occipital_Mid_L, left occipital gyrus. Frontal_Mid_R, right middle frontal gyrus. Frontal_Mid_L, left middle frontal gyrus. Frontal_Sup_R, right dorsolateral frontal gyrus. Cingulum_Ant_R, right anterior cingulate and paracingulate gyrus.

task compared to high PA levels. With age, higher PA levels are associated with prefrontal lobe functioning, thereby reducing the activation levels of related brain regions [49]. Another study found that the active individuals evoked smaller CNV amplitudes in the frontal lobe than the inactive individuals [50]. This suggests that the frontal lobe is a crucial area where PA affects cognitive flexibility in older adults. Results from an ERP study showed that a 6-month open and closed exercise intervention promoted overall electrophysiological effects in the frontal-to-parietal cortices of older adults during a task-switching paradigm and an N-back task [51]. Physical exercise modifies neural function, particularly improving task-switching and working memory abilities in older adults—with pronounced effects in those experiencing cognitive decline.

The activation of the right medial and paracentral cingulate gyrus (BA 6) in physically active young adults was negatively correlated with working memory reaction time. This suggests brain activation in these regions may facilitate quicker cognitive processing during working memory tasks. Importantly, the effects of PA on brain activation patterns appear to be modulated by activity type. For instance, while our study focused on general PA levels, specialized activities like yoga, which combine physical movement with mindfulness, have been shown to elicit distinct neural patterns. Gothe et al. (2018) reported reduced dorsolateral prefrontal cortex activation in yoga practitioners during working memory tasks, potentially reflecting greater neural efficiency through meditative practice. This complements our findings by

demonstrating that how PA is implemented (e.g., aerobic exercise vs. mind-body practices) may determine its cognitive benefits. Similarly, another study found that PA's cognitive benefits are further influenced by lifestyle factors like sleep, with adequate sleep duration enhancing prefrontal activation during working memory tasks [52]. Together, these studies underscore the importance of considering both PA type and complementary lifestyle factors when interpreting neural outcomes. Future research should disentangle these interactions, exploring how different forms of PA, combined with adequate sleep, can synergistically improve cognitive performance in adults.

A significant positive correlation was found between the cognitive flexibility reaction time and activation in the right dorsolateral frontal gyrus (BA 32) and left anterior cingulate and paracingulate gyrus (BA 32) in active older adults. The anterior cingulate cortex is closely associated with cognitive conflict monitoring, error detection, and task-switching control, making it one of the core regions involved in executive function [53]. The right dorsolateral prefrontal cortex, on the other hand, is involved in high-level attentional regulation and flexible adaptation to new rules or goals. This aligns with a meta-analysis indicating that flexibly switched frontoparietal regions form a distributed network involving higher cortical areas, such as dorsolateral prefrontal cortex, anterior cingulate gyrus, and right anterior insula [54]. An intervention study showed that participation in the prefrontal area was significantly enhanced, especially the left superior frontal gyrus and right middle frontal gyrus, after 12 weeks of training. This directly improves the cognitive task-switching ability of the elderly and reduces errors in task-switching [55]. Li et al. reported that a 6-week multimodal intervention improved functional connectivity between the medial prefrontal lobe and medial temporal lobe in resting-state fMRI, which correlated with better task switching and classification fluency [56]. These findings suggest that increased cognitive flexibility is strongly associated with activation of the frontal lobe and other relevant brain areas, especially after targeted physical and cognitive interventions.

It is noteworthy that no significant correlations were observed between brain activation and Flanker task performance in either the young or older groups. This may reflect the Flanker task's relatively higher degree of automatization or lower cognitive load compared to paradigms such as the N-back and task-switching tasks. A previous study showed that the Flanker task elicits limited interindividual variability in behavioral performance, which may constrain its correlation with neural activation [57].

In summary, our study found that PA is associated with more efficient neural processing in young adults, particularly in prefrontal regions critical for working memory and cognitive flexibility. Additionally, while older adults show broader neural recruitment regardless of PA status, higher PA levels correlate with more focal activation patterns, suggesting PA may help optimize compensatory scaffolding. Third, the cognitive benefits of PA appear modulated by both activity type and lifestyle factors like sleep. These findings highlight PA as a potential moderator of age-related neural reorganization, though longitudinal studies are needed to establish causality and disentangle these effects from pre-existing individual differences.

This study has some limitations. Firstly, the cross-sectional design limits more in-depth analyses of the correlation and changes in PA and executive function. The observed brain activation differences may reflect genetic predispositions or baseline cognitive differences rather than PA effects per se. For example, individuals with naturally more efficient neural processing may be both more physically active and show different activation patterns. Future studies can adopt longitudinal and RCT study designs, controlling for genetic and cognitive baseline differences would help clarify these relationships. Secondly, the relatively small sample size may limit the generalization of our findings to the population. The adoption of snowball sampling relies on existing participants to refer others, leading to a non-random sample. This can introduce selection bias, impacting the sample's representativeness of the population. Additionally, PA was measured with a self-reported questionnaire. Participants may overestimate PA levels through recall inaccuracies [58]. Future studies should combine them with accelerometry or heart-rate monitoring for more precise dose-response characterization. Lastly, while Pearson's correlations identified localized PA-related activation differences, future studies could apply independent component analysis to examine large-scale network dynamics. Independent component analysis decomposes fMRI data into functionally coherent networks (e.g., frontoparietal control network) without a priori seed selection [59]. This

data-driven approach may better capture how PA modulates distributed neural circuits supporting executive functions, complementing our task-activated findings.

## Conclusion

Our study yields important insights into how physical activity (PA) relates to neural processing of executive functions across adulthood: (1) Physically inactive older adults showed more widespread brain activation during executive tasks, suggesting either compensatory recruitment or less efficient neural processing; (2) Active young adults demonstrated more focal activation patterns associated with better performance; and (3) Specific prefrontal regions (including the cingulate and dorsolateral frontal gyri) emerged as particularly sensitive to PA effects. These findings suggest that regular PA, particularly activities engaging prefrontal networks, may help maintain efficient brain function during aging. This reinforces PA's potential as a accessible, low-cost strategy to support cognitive vitality in older adults.

## Supporting information

**S1 File.  raw data.**
(PDF)

## Author contributions

**Conceptualization:** Chenglin Zhou, Jing Jin.

**Data curation:** Yingying Wang, Qichen Zhou, Jing Jin.

**Formal analysis:** Yingying Wang, Qichen Zhou, Jing Jin.

**Investigation:** Jing Jin.

**Methodology:** Jing Jin.

**Project administration:** Jing Jin.

**Supervision:** Jing Jin.

**Visualization:** Huiqi SONG, Jie Feng, Yingying Wang, Qichen Zhou.

**Writing – original draft:** Huiqi SONG, Jie Feng, Jing Jin.

**Writing – review & editing:** Huiqi SONG, Jie Feng, Yingying Wang, Qichen Zhou, Chenglin Zhou, Jing Jin.

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
