## [Decision Letter · Decision Letter 0]

PLOS ONE

Dear Dr. Jin,

Thank you for submitting your manuscript to PLOS ONE. After careful consideration, we feel that it has merit but does not fully meet PLOS ONE’s publication criteria as it currently stands. Therefore, we invite you to submit a revised version of the manuscript that addresses the points raised during the review process. As you will find from the individual reviewers' comments below, in particular the following points need clarification:

**Clearer Theoretical Connection** : Reviewers request a stronger, more explicit link among physical activity, executive function, and aging in the Introduction, emphasizing the need to establish a solid theoretical rationale.**Methodological Clarity** : Reviewers note unclear participant criteria (e.g., education requirements, age definitions), discrepancies in participant numbers, and insufficient justification for key thresholds (e.g., 3000 MET-min/week), all pointing to gaps in methodological rigor.**Statistical Transparency** : Reviewrs underscore the importance of clearly detailing statistical procedures and corrections for multiple comparisons, calling for consistent and transparent reporting practices.**Depth of Results Discussion** : Reviewers advocate for more comprehensive reporting of effect sizes, a clearer explanation of the functional significance of activated brain regions, and a stronger tie-in with relevant literature.**Editorial and Limitations** : Reviewers highlight issues with grammar, ask for more direct acknowledgment of study limitations (e.g., cross-sectional design, self-reported PA), and caution against implying causality where only associations exist.

We look forward to receiving your revised manuscript.

Kind regards,

Florian Ph.S Fischmeister

Academic Editor

PLOS ONE

2. In the online submission form, you indicated that [The datasets generated for this study are available on request to the corresponding author.].

Reviewers' comments:

Reviewer's Responses to Questions

**Comments to the Author**

1. Is the manuscript technically sound, and do the data support the conclusions?

Reviewer #1: Yes

Reviewer #2: Partly

Reviewer #3: Yes

2. Has the statistical analysis been performed appropriately and rigorously?

Reviewer #1: Yes

Reviewer #2: I Don't Know

Reviewer #3: Yes

3. Have the authors made all data underlying the findings in their manuscript fully available?

Reviewer #1: Yes

Reviewer #2: Yes

Reviewer #3: Yes

4. Is the manuscript presented in an intelligible fashion and written in standard English?

Reviewer #1: Yes

Reviewer #2: Yes

Reviewer #3: Yes

Reviewer #1: Thank you for the opportunity to review this manuscript for PLOS ONE. I appreciate the chance to contribute to the evaluation of research in this important field. The study provides valuable insights into the relationship between physical activity and brain activation in executive function. The findings contribute to our understanding of cognitive aging and highlight important neural mechanisms underlying physical activity’s impact on cognition. I have provided comments and suggestions to enhance the clarity, depth, and rigor of the manuscript. I hope these recommendations will strengthen the manuscript and further its contribution to the field.

Abstract:

The abstract is well-written, but it could benefit from greater specificity, clarity, and detail in certain areas. Please consider providing more context, statistical details, and explicit comparisons between age groups to strengthen the abstract and make it more informative for readers.

Introduction:

1. The introduction defines executive function broadly but does not explicitly link it to aging and physical activity. While the three core components are well-described, their specific relevance to aging and PA should be emphasized earlier.

2. The demographic statistics on aging are informative but could be more directly tied to the study’s research question. For example, instead of just stating the numbers, explain how the growing aging population increases the urgency for interventions like PA to mitigate cognitive decline.

3. The transition to discussing PA feels abrupt. The introduction should explicitly “connect executive function decline with the potential of PA to mitigate these effects” before presenting the research evidence.

4. The statement "However, the mechanism that explains the relationship between PA and executive function is still not well understood" is vague. Specify which mechanisms (e.g., neuroplasticity, functional connectivity, or neural efficiency) remain unclear. Expand on the mechanisms by adding examples.

5. The introduction mentions that fMRI is underutilized but does not justify why it is a valuable tool for this study. Explain why fMRI provides better insights compared to ERPs or fNIRS and what specific neural mechanisms it can reveal.

6. While the research gap is identified, it needs stronger justification. Specify whether prior studies on PA and executive function lacked neural evidence, were limited to younger adults, or used insufficient methodologies.

7. The objectives are somewhat general. Instead of “analyze the association between brain activation patterns and executive function task performance,” specify which tasks or brain regions are of interest.

8. Some sentences are repetitive or overly verbose. For example, the description of the three core components of executive function could be more concise.

9. Some phrases are overly informal or vague (e.g., "Through decades of research in this field"). The tone should be more precise and scholarly.

Methods

1. The methods section is comprehensive but could benefit from better organization and subheadings to improve readability. For example, the fMRI data acquisition and analysis section is dense and could be broken into smaller subsections (e.g., "Preprocessing," "First-Level Analysis," "Second-Level Analysis").

- Some sentences are overly long and could be simplified for clarity.

2. While the methods are detailed, some key details are missing or unclear, which could hinder reproducibility. For example, the exact version of the software/tools used (e.g., DPARSF, SPM8) should be specified and the rationale for certain parameters (e.g., voxel-significance level, Gaussian kernel size) should be explained.

3. Some methodological choices (e.g., the use of IPAQ-SF, the cutoff for MET-min/week, the exclusion criteria for head motion) lack justification. Providing references or rationale for these decisions would strengthen the section.

Study Design:

1. The use of snowball sampling may introduce selection bias. Consider discussing its limitations and potential impact on generalizability.

2. The criteria are well-defined, but the rationale for some choices is unclear. For example:

- Why was 10 years of education chosen as a cutoff? Is this based on prior literature or specific to the study population?

- The exclusion of participants with hearing, vision, or communication impairments is reasonable, but it should be clarified whether these were self-reported or assessed objectively.

- The inclusion of Han Chinese adults as a criterion should be justified. Was this to control for genetic or cultural factors? If so, this should be explicitly stated. Justify these criteria with references or explanations.

3. The use of the Montreal Cognitive Assessment Basic is appropriate, but the cutoff score (≥19) should be justified with a reference or explanation.

4. The timeline for data collection (October 2021 to December 2021) conflicts with the recruitment period (October 2019 to December 2019). This discrepancy should be clarified.

Measurements

1. The use of IPAQ-SF is appropriate, but the authors should justify the choice of the 3000 MET-min/week cutoff for categorizing participants as active or inactive.

- The reliability and validity of the IPAQ-SF in the studied population should be noted. If validated in a Chinese cohort, citing a relevant study would strengthen credibility.

2. The description of the tasks (Flanker, N-back, Switching) is detailed, but some aspects could be clarified:

- Task Order: The order of tasks (Flanker, N-back, Switching) is mentioned, but the rationale for this order should be explained. Was it randomized or fixed to avoid order effects?

- Practice Sessions: The mention of "thorough practice sessions" is vague. How many practice trials were conducted, and how was proficiency assessed?

- Reaction Time Exclusion Criteria: The exclusion of trials with reaction times <100 ms or >2800 ms is reasonable, but the rationale for these thresholds should be provided.

- Suggestion: Provide more details about task administration, practice sessions, and justification for reaction time thresholds.

Imaging Data Acquisition and Analysis

1. The acquisition parameters are well-described, but the rationale for certain choices (e.g., flip angle, slice thickness) should be explained.

2. The preprocessing steps are clearly outlined, but the version of the DPARSF toolbox should be specified.

- The exclusion criterion for head motion (>2 mm) is mentioned, but the rationale for this threshold should be provided. Additionally, specify how many participants were excluded due to excessive motion.

3. The use of Statistical Parametric Mapping (SPM8) is appropriate, but the version of SPM should be specified. Given that SPM8 is outdated, justify why this version was chosen over more recent alternatives (e.g., SPM12, FSL, AFNI).

- The description of the first- and second-level analyses is clear, but the rationale for the contrast values (e.g., incongruent - congruent for the Flanker task) should be explained.

- The use of Gaussian random field (GRF) theory for multiple comparison correction is appropriate, but the rationale for the chosen significance levels (voxel-level p<0.01, whole-brain p<0.05) should be provided. Specify the cluster-forming threshold used in the GRF correction.

4. The generation of ROIs is described, but the rationale for using a 6 mm-radius sphere should be explained.

5. The correlation analysis between behavioral and imaging metrics is well-described, but the rationale for setting marginal significance at p<0.1 should be explained.

- Ethical approval is mentioned, however, it would be helpful to briefly describe how participant confidentiality and data security were maintained.

- While not typically included in the methods section, a brief acknowledgment of methodological limitations (e.g., cross-sectional design, potential biases in self-reported PA) would strengthen the section.

Results:

This section presents important findings but would benefit from increased clarity, additional statistical reporting (effect sizes, multiple comparisons corrections), and improved discussion of the functional relevance of brain regions.

1. The description of participant inclusion and exclusion could be streamlined. For example, instead of repeating "physically active" multiple times, a summary comparison might make the statistics more readable.

- The phrase "There are 60.0% of participants physically active in the older adults group" should be revised for grammatical accuracy.

2. The results mention p-values, but effect sizes (e.g., Cohen's d, η²) are not reported. Effect sizes are crucial for understanding the magnitude of observed differences, particularly for comparisons of executive function between active and inactive groups.

- Multiple significance tests are reported without mention of corrections for multiple comparisons (e.g., Bonferroni correction or False Discovery Rate). This should be addressed to reduce the risk of Type I errors.

3. The fMRI activation results contain many anatomical terms and Brodmann areas (BA), but the functional significance of these regions is not always clear. A brief description of what these areas are generally associated with (e.g., cognitive control, working memory) would help readers interpret the findings.

- Also, 3.3. Functional activations, inconsistencies exist in the comparisons; some comparisons discuss physically inactive vs. active, while others describe activation differences within one group (e.g., young adults). A clearer structure for each task would improve readability.

4. The correlation results mention no significant correlation in several instances, but it would be useful to discuss whether the non-significant results were expected or whether they suggest that brain activation and executive function may not be strongly linked in this context.

- A key significant correlation (e.g., r = -0.804, p = 0.003 for the 2-back task in young adults) is mentioned but lacks discussion on its practical implications. Why might this correlation be strong? How does it relate to the broader literature on physical activity and cognitive function?

5. Figures and Tables: The text should more explicitly refer to the figures and tables to guide the reader (e.g., "As shown in Table 2,..." instead of "see more details in Table 2").

- If applicable, reporting confidence intervals alongside mean values and p-values in tables would improve the robustness of the statistical findings.

6. Consistency and Accuracy: In section 3.3, there are inconsistencies where "physically inactive" is mistakenly repeated in places where "physically active" should be mentioned (e.g., "activation in the bilateral medial and paracentral cingulate gyrus (BA 24) of the physically inactive group was significantly greater than the physically inactive group"). These should be carefully revised.

- The transition between different tasks (Flanker, N-back, Switching) could be made clearer by adding short summary statements about the key findings at the end of each subsection.

Discussion:

Overall, the discussion presents valuable insights into the effects of PA on brain activation and executive function. Strengthening the logical flow, refining the interpretation of findings, and enhancing comparisons with existing literature will improve the rigor and clarity of this section.

1- The discussion is structured well but would benefit from clearer transitions between sections. Some paragraphs introduce new findings abruptly without linking them to previous discussions. Consider using transitional phrases to improve flow.

- The section should explicitly highlight the main takeaways. A concluding paragraph summarizing key insights would strengthen the discussion.

2- The discussion should elaborate on the theoretical implications of the findings. How do these results contribute to existing models of cognitive aging and PA? For instance, the dedifferentiation hypothesis is mentioned, but its broader implications for aging research could be expanded.

- Some findings are stated without sufficient interpretation. For example, the statement: "Low to moderate levels of PA need greater activation of brain regions than high levels of PA" could be expanded by explaining why this pattern might emerge from a neurophysiological perspective.

3. While the study references prior research, some comparisons are vague. The phrase “this study found increased differences in brain activation across PA levels” should be contextualized within previous findings. How do these differences compare quantitatively or qualitatively with past studies?

- Some cited studies (e.g., yoga practitioners vs. controls) focus on specific types of PA but are not directly linked to the present findings. A stronger rationale for including these comparisons is needed.

4. The discussion acknowledges limitations, but some aspects require further elaboration. For instance, using self-reported PA data is mentioned as a limitation, but what potential biases might this introduce (e.g., recall bias, social desirability bias)?

- The suggestion for future research using the independent component algorithm is valid, but why is this method particularly useful for analyzing executive function and PA? A brief explanation of its advantages over Pearson’s correlation would be beneficial.

5. Some correlations are interpreted in a way that may suggest causality. For example, stating that PA “enhances” prefrontal lobe functioning should be revised to indicate association rather than direct causation due to the cross-sectional nature of the study.

- The study should acknowledge alternative explanations for the observed brain activation differences, such as genetic predispositions or pre-existing cognitive differences that might confound the results.

6. Some sentences are overly complex, reducing readability. For instance, “Exercise-induced neurocognitive changes affect task switching and working memory in older adults, especially with cognitive impairment” could be reworded for clarity.

- There are minor grammatical inconsistencies (e.g., "Besides, PA was measured..." → "Additionally, PA was measured..."). Consider revising for formal academic tone.

Conclusion

The conclusion provides a concise summary of the study’s results but could be expanded to reinforce the study's contributions. Instead of merely restating findings, briefly highlight their broader implications for cognitive aging and PA.

- The statement “inactive older adults needed to activate more brain regions to complete executive function tasks” should specify whether this supports a compensatory mechanism or reflects inefficiency in cognitive processing.

- The conclusion should discuss the relevance of the findings in a broader context. For example, what do these results suggest about PA as a potential intervention for cognitive decline?

- Consider adding a sentence on how these findings align with or challenge existing literature, reinforcing the study’s contribution to the field.

- The conclusion does not mention any limitations, which should be briefly acknowledged to provide a balanced interpretation. Reference issues like the cross-sectional design, self-reported PA measures, or sample size constraints.

- Suggesting potential future research directions, such as longitudinal studies or intervention trials, would strengthen the conclusion.

- The final paragraph is dense with technical terms (e.g., "right medial and paracentral cingulate gyrus (BA 6)"), which may make it less accessible. Consider summarizing key findings in simpler terms before mentioning specific brain regions.

- The conclusion could be more structured, moving from a summary of findings to broader implications and then future research directions.

Reviewer #2: Title: fMRI insights into differential brain activation and physical activity in older adults

Author: Huiqi SONG, Jie FENG, Yingying WANG, Qichen ZHOU, Chenglin ZHOU, Jing JIN

Date of review: 02/21/2025

Strengths of the article

1. Relevance of the subject

The article deals with a subject of major interest in cognitive neuroscience and ageing, namely the impact of physical activity on executive function and brain activation. The research question is well motivated and supported by references.

2. Well-detailed methodology

The study uses well-established tasks to assess executive function (Flanker, N-back, Switching). The use of fMRI to analyse brain activation is relevant and justified.

3. Interesting results

The study shows interesting differences in brain activation depending on the level of physical activity. The finding that physically active adults require less activation to perform executive tasks is consistent with the hypothesis of better cognitive efficiency.

Weaknesses and suggestions for improvement

1. General comments regarding the document

Q1. There are several spelling errors: the term executive functions is plural. It is also written “recreation” instead of “reaction” in several places in the document.

2. Title page

“fMRI insights into differential brain activation and physical activity in older adults”

Q2. Why don't you mention the term “executive functions” in the long title?

3. Abstract

“Methods: This cross-sectional study involved 51 Han Chinese adults (22 young, 31 older) who were categorized as physically active (≥3000 MET-min/week) or inactive (<3000 METmin/ week).”

Q3. Please explicit the term “MET-min/week”.

4. Methods

- Study design and participants

“The inclusion criteria were: (1) Han Chinese adults aged 18 years or older; (2) had more than 10 years of education”

Q4. What was the exact upper age limit in the young group? What were the exact lower and upper age limits in the older group?

Q5. The mean educational level in the older group is 11.10 +/- 1.97. It means that 1 or more participant(s) had less than 10 years of education. How do you explain that discrepancy?

“A total of 51 participants were included, including 22 young adults and 31 older adults.”

Q6. I do not understand this sentence as 22+31 = 53.

Q7. How was the sample size calculated? This information must be indicated according to the Strengthening the Reporting of Observational Studies in Epidemiology (STROBE) statement.

- Imaging data acquisition and analysis

Q8. Why did you not include imaging in the “Measurements” section?

Q9. What corrections for multiple testing were applied? They must be reported in each table.

Q10. How did you select ROIs?

- Statistical methods

Q11. Statistical methods must be described (see the STROBE statement). There is no indication of which statistical tests were used to analyze data other than brain imaging data. Nor is it indicated which dispersion measure is presented in the article: standard deviation or standard error of the mean? I recommend to add a section 2.4. “Statistical methods” or “Statistical analyses”.

5. Results

“Among 52 participants who agreed to participate in this study”

Q12. Why were 51 participants included and 52 participants who agreed to participate?

Q13. What was the mean physical activity level in each group? This must be indicated in the Table 1.

Q14. How did you choose the physical activity cut-off between physically active and physically inactive (3000 MET-min/week)? Is it relevant to choose to use the same cut-off for young and older groups? One may argue that an older person will be considered as active with less activity than a younger person.

Q15. Results must indicate the statistics value (for example: t value if a Student t test has been used or the F value if an ANOVA has been performed) wherever necessary. P values are not sufficient.

6. Table 1.

“Data are presented as mean (SD) or number (percentage).”

Q16. I do not understand this sentence. It seems that some lines indicate means +/- SD and that other lines indicate the number of participants with percentages in brackets. Is it correct?

Reviewer #3: The study is well-structured and explores an important topic. However, minor revisions are needed to clarify the justification for the 3000 MET-min/week threshold and ensure greater scientific accuracy and clarity.

**Do you want your identity to be public for this peer review?** For information about this choice, including consent withdrawal, please see our Privacy Policy

Reviewer #1: **Yes: ** Marjan Hosseini

Reviewer #2: No

Reviewer #3: No

---

## [Author Response · Author response to Decision Letter 1]

30 Apr 2025

Response to reviewers

Review Comments to the Author

Reviewer #1: Thank you for the opportunity to review this manuscript for PLOS ONE. I appreciate the chance to contribute to the evaluation of research in this important field. The study provides valuable insights into the relationship between physical activity and brain activation in executive function. The findings contribute to our understanding of cognitive aging and highlight important neural mechanisms underlying physical activity’s impact on cognition. I have provided comments and suggestions to enhance the clarity, depth, and rigor of the manuscript. I hope these recommendations will strengthen the manuscript and further its contribution to the field.

Response: Thank you. We have revised the manuscript according to your valuable comments.

Abstract:

The abstract is well-written, but it could benefit from greater specificity, clarity, and detail in certain areas. Please consider providing more context, statistical details, and explicit comparisons between age groups to strengthen the abstract and make it more informative for readers.

Response: Thanks for your suggestion. We have revised the abstract to make it more informative for readers. We have now revised the abstract as follows:

“Background: Executive function is vital for cognitive health, particularly in older adults, where declines can lead to an increased risk of cognitive impairment. Physical activity (PA) has been linked to improvements in executive function, yet the underlying mechanisms remain poorly understood.

Methods: This cross-sectional study involved 41 Chinese adults (21 young: 23.0 ± 2.12 years; 20 older: 63.30 ± 2.36 years) who were categorized as physically active (≥3000 metabolic equivalent (MET)-min/week) or inactive (<3000 MET-min/week). Participants performed fMRI while completing executive function tasks (Flanker, N-back, Switching). Brain activation patterns were analyzed using Statistical Parametric Mapping (SPM), with significance thresholds set at p< 0.01 (voxel-level) and p< 0.05 (whole-brain corrected).

Results: Physically active older adults showed significantly better accuracy and faster reaction times on the Flanker task than inactive peers. In young adults, those who were inactive exhibited greater activation in prefrontal regions during executive tasks. No significant differences in brain activation were found in older adults for these tasks. Additionally, activation in the right medial/paracentral cingulate gyrus (BA 6) negatively correlated with working memory reaction times in active young adults (r= −0.804, p< 0.05), whereas cognitive flexibility in active older adults positively correlated with activation in the right dorsolateral frontal gyrus (BA 32; r = 0.589, p< 0.05).

Conclusion: Active older adults require less brain activation to perform executive function tasks, suggesting enhanced cognitive efficiency. In contrast, young adults showed different patterns of brain activation, indicating potential compensatory mechanisms. These results underscore PA’s role in optimizing age-specific cognitive strategies and underscore the need for longitudinal research to clarify causality.”

Introduction:

1. The introduction defines executive function broadly but does not explicitly link it to aging and physical activity. While the three core components are well-described, their specific relevance to aging and PA should be emphasized earlier.

Response: Thanks for your comment. Combined with other comments, we have now re-organized the sequence of the Introduction part and specified the relevance of three core components to aging and PA, with one additional reference supporting this:

“Similarly, a review of 25 randomized controlled trials involving adults over 60 years old found that PA positively enhanced inhibitory control, working memory, and cognitive flexibility [15].”

Reference

15. Xiong J, Ye M, Wang L, Zheng G. Effects of physical exercise on executive function in cognitively healthy older adults: A systematic review and meta-analysis of randomized controlled trials: Physical exercise for executive function. International Journal of Nursing Studies. 2021;114:103810. doi: https://doi.org/10.1016/j.ijnurstu.2020.103810.

2. The demographic statistics on aging are informative but could be more directly tied to the study’s research question. For example, instead of just stating the numbers, explain how the growing aging population increases the urgency for interventions like PA to mitigate cognitive decline.

Response: Thanks for your suggestions. Combined with other comments, we have restructured the Introduction and revised the sentence as follows:

“This demographic transformation has intensified age-related public health challenges, particularly the decline in cognitive functions such as executive function, which is strongly associated with an increased risk of mild cognitive impairment and dementia [2], as well as deterioration in brain structure and function [3].”

3. The transition to discussing PA feels abrupt. The introduction should explicitly “connect executive function decline with the potential of PA to mitigate these effects” before presenting the research evidence.

Response: Thanks for your suggestion. We have now added one more sentence to link these two paragraphs, with one additional reference.

“Given the profound implications of executive function decline on daily functioning and independence [8], identifying strategies to mitigate these effects has become a crucial area of research.”

Reference

8. Pérez Palmer N, Trejo Ortega B, Joshi P. Cognitive Impairment in Older Adults: Epidemiology, Diagnosis, and Treatment. Psychiatric Clinics. 2022;45(4):639-61. doi: 10.1016/j.psc.2022.07.010.

4. The statement "However, the mechanism that explains the relationship between PA and executive function is still not well understood" is vague. Specify which mechanisms (e.g., neuroplasticity, functional connectivity, or neural efficiency) remain unclear. Expand on the mechanisms by adding examples.

Response: Thanks for raising this issue. Combined with other comments, we have now specified and expanded the mechanism, with three additional references supporting our rationale. The newly added sentences are as follows:

“However, the mechanism that explain the relationship between PA and executive function is still not well understood, particularly regarding how PA influences functional connectivity and neural efficiency.”

“While these findings provide valuable insights, they are limited in their ability to comprehensively map whole-brain mechanisms or clarify how PA enhances functional connectivity and neuroplasticity across distributed neural networks.”

“Functional magnetic resonance imaging (fMRI) is often considered as the gold standard for the assessment of brain activity, and offers superior spatial resolution compared to methodologies such as ERPs and fNIRS [19], enabling precise localization of neural activity across distributed brain networks critical for executive function. Unlike ERPs, which excel in temporal resolution but lack spatial precision, and fNIRS, which is limited to cortical surface measurements, fMRI provides comprehensive mapping of both localized and network-level brain activity. This capability allows for the assessment of task-evoked activation and resting-state connectivity, offering unique insights into how PA modulates neural efficiency, plasticity, and compensatory mechanisms, particularly in aging populations. However, prior research on PA and executive function has predominantly relied on ERPs and fNIRS [20, 21], which are constrained in their ability to examine whole-brain neural mechanisms. Additionally, many studies have focused on younger adults or lacked the spatial resolution necessary to investigate age-related neural adaptations. fMRI addresses these limitations by providing a robust framework to explore PA-related neural changes across the lifespan, filling critical gaps in understanding how PA influences executive function in older adults.”

Reference

19. Glover GH. Overview of functional magnetic resonance imaging. Neurosurg Clin N Am. 2011;22(2):133-9, vii. doi: 10.1016/j.nec.2010.11.001. PubMed PMID: 21435566; PubMed Central PMCID: PMCPMC3073717.

20. Chainay H, Joubert C, Massol S. Behavioural and ERP Effects of Cognitive and Combined Cognitive and Physical Training on Working Memory and Executive Function in Healthy Older Adults. Adv Cogn Psychol. 2021;17(1):58-69. Epub 20210406. doi: 10.5709/acp-0317-y. PubMed PMID: 35003404; PubMed Central PMCID: PMCPMC8720364.

21. Shen Q-Q, Hou J-M, Xia T, Zhang J-Y, Wang D-L, Yang Y, et al. Exercise promotes brain health: a systematic review of fNIRS studies. Frontiers in Psychology. 2024;Volume 15 - 2024. doi: 10.3389/fpsyg.2024.1327822.

5. The introduction mentions that fMRI is underutilized but does not justify why it is a valuable tool for this study. Explain why fMRI provides better insights compared to ERPs or fNIRS and what specific neural mechanisms it can reveal.

Response: Thanks for your suggestion. Combined with other comments, we have now provided rationale for using fMRI, with three additional references supporting our rationale. The newly added sentences are as follows:

“Functional magnetic resonance imaging (fMRI) is often considered as the gold standard for the assessment of brain activity, and offers superior spatial resolution compared to methodologies such as ERPs and fNIRS [19], enabling precise localization of neural activity across distributed brain networks critical for executive function. Unlike ERPs, which excel in temporal resolution but lack spatial precision, and fNIRS, which is limited to cortical surface measurements, fMRI provides comprehensive mapping of both localized and network-level brain activity. This capability allows for the assessment of task-evoked activation and resting-state connectivity, offering unique insights into how PA modulates neural efficiency, plasticity, and compensatory mechanisms, particularly in aging populations. However, prior research on PA and executive function has predominantly relied on ERPs and fNIRS [20, 21], which are constrained in their ability to examine whole-brain neural mechanisms. Additionally, many studies have focused on younger adults or lacked the spatial resolution necessary to investigate age-related neural adaptations. fMRI addresses these limitations by providing a robust framework to explore PA-related neural changes across the lifespan, filling critical gaps in understanding how PA influences executive function in older adults.”

Reference

19. Glover GH. Overview of functional magnetic resonance imaging. Neurosurg Clin N Am. 2011;22(2):133-9, vii. doi: 10.1016/j.nec.2010.11.001. PubMed PMID: 21435566; PubMed Central PMCID: PMCPMC3073717.

20. Chainay H, Joubert C, Massol S. Behavioural and ERP Effects of Cognitive and Combined Cognitive and Physical Training on Working Memory and Executive Function in Healthy Older Adults. Adv Cogn Psychol. 2021;17(1):58-69. Epub 20210406. doi: 10.5709/acp-0317-y. PubMed PMID: 35003404; PubMed Central PMCID: PMCPMC8720364.

21. Shen Q-Q, Hou J-M, Xia T, Zhang J-Y, Wang D-L, Yang Y, et al. Exercise promotes brain health: a systematic review of fNIRS studies. Frontiers in Psychology. 2024;Volume 15 - 2024. doi: 10.3389/fpsyg.2024.1327822.

6. While the research gap is identified, it needs stronger justification. Specify whether prior studies on PA and executive function lacked neural evidence, were limited to younger adults, or used insufficient methodologies.

Response: Thanks for your suggestion. We have now provided more justification, with two additional references supporting our rationale. The newly added sentences are as follows:

“However, prior research on PA and executive function has predominantly relied on ERPs and fNIRS [20, 21], which are constrained in their ability to examine whole-brain neural mechanisms. Additionally, many studies have focused on younger adults or lacked the spatial resolution necessary to investigate age-related neural adaptations. fMRI addresses these limitations by providing a robust framework to explore PA-related neural changes across the lifespan, filling critical gaps in understanding how PA influences executive function in older adults.”

Reference

20. Chainay H, Joubert C, Massol S. Behavioural and ERP Effects of Cognitive and Combined Cognitive and Physical Training on Working Memory and Executive Function in Healthy Older Adults. Adv Cogn Psychol. 2021;17(1):58-69. Epub 20210406. doi: 10.5709/acp-0317-y. PubMed PMID: 35003404; PubMed Central PMCID: PMCPMC8720364.

21. Shen Q-Q, Hou J-M, Xia T, Zhang J-Y, Wang D-L, Yang Y, et al. Exercise promotes brain health: a systematic review of fNIRS studies. Frontiers in Psychology. 2024;Volume 15 - 2024. doi: 10.3389/fpsyg.2024.1327822.

7. The objectives are somewhat general. Instead of “analyze the association between brain activation patterns and executive function task performance,” specify which tasks or brain regions are of interest.

Response: We have now revised the sentence as follows:

“explore the association between brain activation patterns and performance in inhibitory control, working memory, and cognitive flexibility tasks.”

8. Some sentences are repetitive or overly verbose. For example, the description of the three core components of executive function could be more concise.

Response: Thanks for raising this concern. We have now revised this sentence as follows:

“These interconnected components serve distinct yet complementary functions: inhibitory control suppresses irrelevant stimuli, working memory stores and manipulates information, and cognitive flexibility adapts to shifting demands.”

9. Some phrases are overly informal or vague (e.g., "Through decades of research in this field"). The tone should be more precise and scholarly.

Response: Thanks for raising this concern. We have now revised this sentence as follows:

“Extensive empirical research conducted over several decades has established a consensus that executive function comprises three core fundamental components: inhibitory control, working memory, and cognitive flexibility.”

Methods

1. The methods section is comprehensive but could benefit from better organization and subheadings to improve readability. For example, the fMRI data acquisition and analysis section is dense and could be broken into smaller subsections (e.g., "Preprocessing," "First-Level Analysis," "Second-Level Analysis").

- Some sentences are overly long and could be simplified for clarity.

Response: Thanks for your suggestion. We have reorganized the method section and added subheadings to improve readability. We also simplified some sentences for clarity.

2. While the methods are detailed, some key details are missing or unclear, which could hinder reproducibility. For example, the exact version of the software/tools used (e.g., DPARSF, SPM8) should be specified and the rationale for certain parameters (e.g., voxel-significance level, Gaussian kernel size) should be explained.

Response: Thanks for your suggestion. We have added the exact version of the software/tools used. We have now revised this sentence as follows:

“Imaging data were preprocessed using the Data Processing Assistant for Resting-State fMRI (DPARSF) toolbox (http://rfmri.org/DPARSF) [35]”

“After preprocessing, the data were analyzed using Statistical Parametric Mapping (SPM8) (Wellcome Centre for Human Neuroimaging, London, UK, http://www.fil.ion.ucl.ac.uk/spm).”

The rationale for key parameter selection as follows:

The selection of Gaussian Random Field (GRF) parameters was carefully optimized through several considerations: (1) For voxel-level thresholding, we adopted p<0.01 as this provides an optimal balance between Type I and Type II error rates; (2) The cluster-level threshold of p<0.05 (FWE-corrected) was implemented to maintain strict family-wise error control while preserving reasonable sensitivity; (3) Cluster formation employed a two-stage approach using an initial contiguous voxel threshold of p<0.001 (uncorrected) for cluster definition followed by GRF-based cluster extent thresholding. This comprehensive approach ensures robust statistical c

---

## [Decision Letter · Decision Letter 1]

Dear Dr. Jin,

Thank you for submitting your manuscript to PLOS ONE. After careful consideration, we feel that it has merit but does not fully meet PLOS ONE’s publication criteria as it currently stands. Therefore, we invite you to submit a revised version of the manuscript clarifying and justifying the chosen 3000 MET-min/week threshold.

We look forward to receiving your revised manuscript.

Kind regards,

Florian Ph.S Fischmeister

Academic Editor

PLOS ONE

Journal Requirements:

Reviewers' comments:

Reviewer's Responses to Questions

**Comments to the Author**

Reviewer #2: All comments have been addressed

Reviewer #3: (No Response)

2. Is the manuscript technically sound, and do the data support the conclusions?

Reviewer #2: Yes

Reviewer #3: Yes

3. Has the statistical analysis been performed appropriately and rigorously?

Reviewer #2: Yes

Reviewer #3: Yes

4. Have the authors made all data underlying the findings in their manuscript fully available?

Reviewer #2: No

Reviewer #3: Yes

5. Is the manuscript presented in an intelligible fashion and written in standard English?

Reviewer #2: Yes

Reviewer #3: Yes

Reviewer #2: I would like to thank the authors for answering all our questions and comments, and for modifying their article accordingly. I have no further comments.

Reviewer #3: (No Response)

**Do you want your identity to be public for this peer review?** For information about this choice, including consent withdrawal, please see our Privacy Policy

Reviewer #2: No

Reviewer #3: No

---

## [Author Response · Author response to Decision Letter 2]

4 Jun 2025

Review Comments to the Author

The authors have provided references to support their use of the 3000 MET-min/week threshold (Andersen et al., 2021; Macfarlane et al., 2007). While these sources confirm the general applicability of the IPAQ-SF and its validity in certain populations, they do not sufficiently explain why this particular cutoff was chosen in the context of cognitive health or brain activation.

This threshold is not commonly used in neuroimaging or cognitive neuroscience research. Therefore, its selection in this study should be either:

- clearly justified with reference to prior work or guidelines related to brain health (e.g., WHO, ACSM), or

- explicitly presented as a novel, hypothesis-driven approach to examine possible neurocognitive associations.

Providing this clarification will strengthen the conceptual alignment between the study’s aims and its methodological framework.

Response: Thanks for your valuable suggestions. We have added justification of the 3000 MET-min/week threshold. We have now revised this sentence as follows:

“Participants were classified into the physically active group with PA achieving ≥3000 MET-min/week and the physically inactive group with PA achieving <3000 MET-min/week. This threshold aligns with the high activity category defined in the IPAQ scoring protocol [24]. The higher threshold is selected based on evidence that found PA was positively associated with brain health in older adults [26].”

Reference

24. Craig CL, Marshall AL, Sjöström M, Bauman AE, Booth ML, Ainsworth BE, et al. International physical activity questionnaire: 12-country reliability and validity.

Medicine & science in sports & exercise. 2003;35(8):1381-95.

26. La Hood A, Moran C, Than S, Lu A, Collyer TA, Beare R, et al. Associations between physical activity and brain structure in a large community cohort. Scientific Reports. 2025;15(1):18896. doi: 10.1038/s41598-025-04010-7.

---

## [Editor Report · Decision Letter 2]

fMRI insights into differential brain activation, executive function, and physical activity in older adults

PONE-D-24-57669R2

Dear Dr. Jin,

We’re pleased to inform you that your manuscript has been judged scientifically suitable for publication and will be formally accepted for publication once it meets all outstanding technical requirements.

Kind regards,

Florian Ph.S Fischmeister

Academic Editor

PLOS ONE
---

## [Editor Report · Acceptance letter]

PONE-D-24-57669R2

PLOS ONE

Dear Dr. Jin,

I'm pleased to inform you that your manuscript has been deemed suitable for publication in PLOS ONE. Congratulations! Your manuscript is now being handed over to our production team.

Kind regards,

on behalf of

Mag. Dr. Florian Ph.S Fischmeister

Academic Editor

PLOS ONE